# Single-Port Versus Multi-Port Robotic Radical Prostatectomy in Elderly Patients

**DOI:** 10.3390/cancers17111857

**Published:** 2025-05-31

**Authors:** Fabio Maria Valenzi, Valerio Santarelli, Giulio Avesani, Muhannad Aljoulani, Hakan Bahadir Haberal, Juan R. Torres Anguiano, Luca Alfredo Morgantini, Ruben Sauer Calvo, Arianna Biasatti, Andrea Fuschi, Antonio Luigi Pastore, Simone Crivellaro

**Affiliations:** 1Department of Urology, University of Illinois at Chicago, Chicago, IL 60607, USAbahadirhaberal@gmail.com (H.B.H.); jtorre80@uic.edu (J.R.T.A.);; 2Urology Unit, Department of Medico-Surgical Sciences and Biotechnologies, Faculty of Pharmacy and Medicine, Sapienza University of Rome, 04100 Latina, Italy; 3Department of Maternal-Infant and Urological Sciences, Sapienza University of Rome, Umberto I Hospital, 00161 Rome, Italy; 4Division of Oncology/Unit of Urology, Urological Research Institute, IRCCS Ospedale San Raffaele, 20132 Milan, Italy; 5Department of Biomedical Sciences, Humanitas University, 20072 Milan, Italy; 6Urologic Clinic, Department of Medicine, Surgery and Health Sciences, University of Trieste, 34127 Trieste, Italy

**Keywords:** robotic surgery, prostate cancer, single port, elderly, postoperative complications

## Abstract

This study explores the single-port approach to performing radical prostatectomy and compares it to the traditional multi-port method. Furthermore, it specifically focuses on perioperative outcomes in elderly patients between these two robotic approaches. Prostate cancer is common in elderly men, and surgical treatment can be challenging due to age-related risks and slower recovery. The aim of this research is to understand if the single-port approach impacts postoperative complications and perioperative outcomes in older patients. The results of the study showed that elderly patients undergoing single-port radical prostatectomy had fewer short-term complications compared to the multi-port approach, with outcomes significantly correlated with the robotic technique. These findings suggest that single-port robotic surgery is a safe and effective option for elderly patients. This might help guide surgeons in choosing the best surgical approach for older patients undergoing radical prostatectomy.

## 1. Introduction

Prostate cancer (PCa) is mostly diagnosed in patients over the age of 65, with a median age of 66 years in Western countries. Elderly patients often present with multiple comorbidities and reduced biological reserves, which makes them particularly suited for tailored treatment strategies [1,2]. Both radical prostatectomy (RP) and radiotherapy are viable local treatment options for older patients, regardless of the cancer risk stratification [3,4]. While each approach is associated with distinct side effects and complications [5], Kesavan et al. reported a higher rate of cancer-related readmissions among elderly patients treated with radiotherapy compared to those undergoing RP [6]. However, according to the European Association of Urology guidelines, treatment decisions should not be based solely on chronological age. Instead, factors such as life expectancy, health status, frailty and comorbidity must be considered. In this context, the physician-–patient relationship plays a crucial role in shared decision-making [7,8]. Consequently, it is noteworthy that in the United States, only 41% of patients over the age of 75 years receive therapeutic treatment for PCa [9].

Regarding RP, there is still a lack of consensus on the impact of age on both functional and oncological outcomes in elderly patients. This knowledge gap primarily stems from the challenges in enrolling this population in large randomized clinical trials [3,10]. With respect to the multi-port robot-assisted (MP-RARP) approach, several studies have investigated its impact across different age groups, though their findings have been inconsistent. For example, Preisser et al. demonstrated that early postoperative complications following both open and MP-RARP procedures were associated with increasing age [11]. By contrast, Togashi et al. reported higher patient satisfaction scores among older individuals at one year of follow-up [12].

Advances in surgical technology have led to the development of the Da Vinci Surgical System, which has gained widespread recognition for its effectiveness in performing various urologic procedures, including RARP [13,14]. The most recent innovation is the Da Vinci Single-Port system (SP-RARP) (Intuitive Surgical Inc., Sunnyvale, CA, USA) [15]. To date, only one study has specifically examined the impact of SP-RARP in elderly patients, concluding that this approach is both feasible and capable of achieving acceptable oncological and functional outcomes in older patients [16]. However, no studies have yet directly compared the MP and SP approaches focusing on elderly patients.

The present study aims to evaluate the differences between patients undergoing MP-RARP and those undergoing SP-RARP, stratified by age, with a particular focus on intraoperative and perioperative outcomes.

## 2. Materials and Methods

This retrospective study included patients diagnosed with localized PCa who underwent SP- or MP-RARP at the University of Illinois at Chicago between January 2018 and December 2023. Data were collected through the electronic database of hospital records (Epic system^®^, Verona, WI, USA). All procedures were performed by four different surgeons using the same perioperative setting during the study period. Each of the participating surgeons had a high level of experience with robotic surgery. Patients were staged using a risk-based approach to assess the extent of the disease, including the presence of nodal involvement or distant metastasis. Those with locally advanced or metastatic PCa were excluded from the study. Lymphadenectomy was performed according to preoperative imaging findings and validated nomograms. Furthermore, patients who underwent salvage prostatectomy or any other concomitant surgical procedures were also excluded from the study. To facilitate the comparative evaluation, patients were stratified into two age-based cohorts, as performed by Haberal et al. [16]: Group A (patients younger than 65 years) and Group B (patients aged 65 or older).

This study was conducted in accordance with the Declaration of Helsinki, and it was approved by the Institutional Review Board (IRB: 2017-0152). All enrolled patients signed informed consent for both the surgical procedures and the collection of the clinical data for research purposes.

### 2.1. Preoperative Evaluation

Different demographic and clinical characteristics were collected such as age, body mass index (BMI), American Society of Anesthesiologists (ASA) score, Charlson Comorbidity Index (CCI), race, substance abuse, smoking status, history of hypertension, use of hypertensive medications and use of anticoagulation or antiplatelet therapy. Regarding PCa-specific parameters, the following preoperative data were collected: PSA, D’Amico risk classification, International Society of Urological Pathology (ISUP) grade group and clinical T and N stage.

### 2.2. Intraoperative and Postoperative Evaluation

Among the intraoperative and postoperative outcomes, the following data were also collected: surgical approach (transperitoneal or extraperitoneal), operative time, lymphadenectomy, estimated intraoperative blood loss (EBL), intraoperative complication, 30- and 90-days post-op complications according to Clavien–Dindo classification, length of stay (LOS), same-day discharge (SDD) procedures, postoperative nausea and vomiting (PONV), ISUP grade group, pathological T and N stage, positive surgical margins and biochemical recurrence, defined as serum PSA level ≥ 0.2 ng/mL on two consecutive measurements during the follow-up. SDD procedures were defined as those in which patients were discharged within 24 h from hospitalization, after restoring mobility and verifying stable vital signs.

### 2.3. Surgical Procedure

For patients undergoing MP-RARP, a transperitoneal approach was employed, utilizing four robotic arms and two assistant ports. Pneumoperitoneum was induced using a Verres needle or, alternatively, via a mini-open technique when necessary. All transperitoneal procedures were conducted with the patients with 25° Trendelenburg position. Following the robotic docking, the Retzius space was developed through dissection of the umbilical ligaments and bladder peritoneum. RP was then performed using a retropubic approach. For the SP-RARP, either transperitoneal or extraperitoneal approach was used based on surgeon’s preference. The transperitoneal approach followed the same steps as the MP-RARP. Conversely, the extraperitoneal approach was conducted through a 3 cm horizontal incision, positioned approximately four fingerbreadths above the pubic bone, allowing direct access to the Retzius space. A small access port was inserted to allow the dock of the SP robot. The subsequent steps of RP mirrored those employed as previously stated for the MP procedure. Notably, no assistant ports were utilized during SP procedures.

### 2.4. Endpoint

The primary endpoint of the study was to evaluate differences in the rates of 30-day and 90-day postoperative complications in elderly patients undergoing MP versus SP procedures. The secondary objective of the study was to assess differences in length of stay between elderly and non-elderly patients.

### 2.5. Statistical Analysis

Statistical analysis was performed using R software (version 4.4.1; R Foundation for Statistical Computing, Vienna, Austria). A two-sided *p*-value < 0.05 was considered statistically significant. Shapiro–Wilk test was used to analyze the distribution of the variables. Continuous variables were expressed as median with interquartile range (IQR), while categorical variables were expressed as frequencies and percentages (%). Comparison between continuous variables was performed using Mann–Whitney U test, while categorical variables were compared with chi-squared test. Biochemical recurrence-free survival was evaluated using the Kaplan–Meier method, and differences between surgical approaches (MP vs. SP) were assessed with the log-rank test. Simple logistic regression was used to analyze the variables correlated with postoperative complications at 30 days and 90 days in patients of Group B.

## 3. Results

Data from a total of 338 patients were retrospectively collected, including 153 who underwent MP-RARP and 185 who underwent SP-RARP. Preoperative patient characteristics are summarized in Table 1.

The CCI was significantly higher in patients undergoing the SP procedure than those undergoing the MP procedure in both Group A (4 (2) vs. 3 (2), respectively; *p* < 0.001) and Group B (5 (1) vs. 5 (1), respectively; *p* < 0.001. Figure 1). As expected, CCI was also significantly higher in older patients, both in the MP group (5 (1) vs. 3 (2); *p* < 0.001) and in the SP group (5 (1) vs. 4 (2); *p* < 0.001). Regarding BMI, younger patients in the MP group exhibited significantly higher values compared to older patients (29.9 (5.58) vs. 29.9 (7.54), respectively; *p* = 0.025), while no significant differences were observed in other comparisons. Smoking prevalence was significantly higher among MP patients in Group A (64 (64.6%) vs. 37 (35.9%); *p* < 0.001). Additionally, the use of antiplatelet or anticoagulation therapy was significantly more common in older patients in both the SP group (39 (47.6%) vs. 26 (25.2%); *p* = 0.002) and the MP group (30 (55.6%) vs. 32 (32.3%); *p* = 0.005).

Surgical treatment was successfully completed in all cases. Intraoperative and postoperative outcomes are summarized in Table 2. All MP procedures were performed using a transperitoneal approach (100%), while for the SP procedures, 36 (35%) had transperitoneal access in Group A and 30 (36.6%) in Group B underwent a transperitoneal approach, whereas 67 (65%) in Group A and 52 (63.4%) in Group B underwent an extraperitoneal approach. Operation time was significantly shorter in SP procedures compared to MP procedures in both groups (Group A: 241 (56.0) vs. 288 (83.5), respectively; *p* < 0.001; Group B: 256 (72.3) vs. 290 (74.0), respectively; *p* < 0.001). However, no significant differences were observed within the MP or SP groups when comparing younger and older patients (MP: 288 (83.5) vs. 290 (74.0), respectively; *p* = 0.084; SP: 241 (56.0) vs. 256 (72.3), respectively; *p* = 0.098). Lymphadenectomy was more frequently performed in patients undergoing MP procedures in both Group A (75 (75.8%) vs. 63 (61.2%); *p* = 0.026) and Group B (49 (90.7%) vs. 49 (59.8%); *p* < 0.001). Furthermore, within the MP group, older patients were significantly more likely to undergo lymphadenectomy than younger patients (49 (90.7%) vs. 75 (75.8%); *p* = 0.024). Regarding EBL, a significant difference was observed only in Group A, with a lower EBL in the SP cohort compared to MP patients (100 (150) vs. 150 (150), respectively; *p* = 0.008). No statistically significant differences were observed in intraoperative complication rates across groups. In Group A, one intraoperative complication occurred in the MP group (rectal injury), while two complications were reported in the SP Group (rectal injury and bladder opening). In Group B, two intraoperative complications occurred in the MP group (one rectal injury and one conversion to open surgery due to hemodynamic instability in the Trendelenburg position), while six complications were observed in the SP group (one rectal injury, one bladder opening, one obturator nerve injury and two cases of persistent pneumoperitoneum). All intraoperative complications were successfully managed intraoperatively.

LOS was significantly shorter for SP patients in Group B compared to MP patients (18 (20.8) vs. 35 (17), respectively; *p* = 0.002), with a mean hospitalization period of less than 24 h. Furthermore, SDD was achieved significantly more frequently for the SP procedure than for the MP procedure in both Group A (63 (61.2%) vs. 21 (21.2%); *p* < 0.00) and Group B (45 (54.9%) vs. 9 (16.7%); *p* < 0.001), with no significant differences across age groups for both procedures (MP: 21 (21.2%) vs. 9 (16.7%); *p* = 0.643; SP: 63 (61.2%) vs. 45 (54.9%); *p* = 0.477). PONV was also significantly lower in SP patients in Group B compared to MP patients (9 (10.9%) vs. 13 (24.1%); *p*= 0.040). At 30 days postoperation, MP patients in Group B had a significantly higher rate of complications compared to SP patients (19 (35.2%) vs. 13 (15.9%); *p* = 0.009) (Figure 2). Postoperative complications in Group B classified as Clavien–Dindo grade 3 included three cases of paralytic ileus requiring nasogastric decompression and two lymphoceles requiring percutaneous drainage in the MP group, while in the SP group, two lymphoceles required percutaneous drainage. One patient in the MP group died due to postoperative urosepsis (Clavien–Dindo grade 5). At 90 days postoperation, complications remained significantly more frequent among MP patients in Group B (21 (38.9%) vs. 14 (17.1%); *p* = 0.004) (Figure 3). However, no significant differences were observed within the MP or SP groups when comparing younger and older patients regarding postoperative complications at both 30 and 90 days. Furthermore, a lower incidence of biochemical recurrence was observed in SP patients in Group B compared to MP patients (8 (9.8%) vs. 12 (22.2%). When analyzed using a time-dependent method, the recurrence-free survival was significantly longer for SP patients, as shown by the Kaplan–Meier analysis (log-rank test, *p* = 0.040).

To identify the factors most strongly associated with postoperative complications, a multivariable logistic regression analysis was performed for postoperative complications at 30 days (Table 3) and 90 days (Table 4) in older patients (Group B). The surgical approach was the only variable that remained statistically significant for postoperative complications at both 30 days and 90 days. Specifically, the SP approach was associated with a significantly lower risk of complications compared to the MP procedure in terms of postoperative complications at both 30 days (Odds ratio: 0.41; 95% CI: 0.15, 0.97; *p* value = 0.027) and 90 days (Odds ratio: 0.38; 95% CI: 0.17, 0.88; *p* value = 0.024). LOS was also statistically correlated with postoperative complications at only 30 days (Odds ratio: 1.02; 95% CI: 1.01, 1.13; *p* value = 0.046), even though this was close to the threshold for statistical significance.

## 4. Discussion

PCa is the most common solid malignancy among men worldwide and the second most prevalent cancer in the male population [17,18]. Most cases are diagnosed in an early stage, rather than in an advanced stage, and active treatment is recommended, with RARP being one of the most commonly performed surgical procedures [19,20]. Age is a well-established risk factor for the development of PCa, and as life expectancy continues to rise, particularly in Western countries, the incidence of PCa diagnoses is expected to increase, especially among older patients [21,22,23]. Furthermore, the prevalence of PCa is particularly high in Western countries, with incidence rates notably higher among patients aged over 65 years [24,25]. Nonetheless, older age is also a risk factor for worse postoperative outcomes in patients undergoing surgical treatment [21,26]. Advances in surgical technology have introduced new minimally invasive techniques designed to reduce the impact of surgery on patients. Among these, SP robotic surgery has gained increasing popularity worldwide [27,28].

This study aims to evaluate the differences between MP-RARP and SP-RARP, particularly focusing on older patients. Haberal et al. previously demonstrated the safety and feasibility of performing SP-RARP in elderly patients; however, the study did not include any comparison between different robotic platforms [16]. As anticipated, in our cohort, elderly patients exhibited a higher CCI and greater frequencies of antiplatelet and anticoagulation therapy, which highlights the increased frailty of this population compared to younger patients. By contrast, no significant differences were found regarding ASA score or prevalence of hypertension diseases across ages. Preoperative oncological assessment, including D’Amico risk classification, preoperative PSA, ISUP grade group and clinical T and N stages, did not reveal significant differences between patients in Group A and Group B or between those undergoing SP and MP procedures within each age group.

Patients undergoing SP procedures in Group B had a higher CCI compared to those in the same group who underwent MP, which suggests that the SP cohort was frailer at baseline. Apart from this, the population appeared relatively homogeneous.

The operative time was significantly shorter for SP procedures than for MP procedures. This may be attributed to a higher percentage of lymphadenectomies performed during MP procedures, which could increase the operative time, but also to the single-port extraperitoneal approach, which allows for a shorter time to position the patient and to gain access and for skipping some surgical steps (i.e., dropping of the bladder). A statistically significant difference in postoperative complications at 30 and 90 days was observed in favor of the MP procedures in Group B. Using logistic regression, the only statistically significant factor correlated with both types of complications was the kind of procedure. Specifically, SP procedures were associated with a lower incidence of complications compared to MP procedures, with a total reduction of 59% at 30 days and of 62% at 90 days. This suggests that SP may be a protective factor for these events in elderly patients.

One possible explanation for this finding is that the SP procedure, with its flexible instruments and ability to operate in a narrower space, causes less damage to the surrounding organs and tissues. Although this advantage is particularly evident in the extraperitoneal approach, which was used only in the SP group, the approach itself was not correlated with postoperative complications in elderly patients. Nonetheless, all MP procedures were performed with a transperitoneal approach, which may have influenced the statistical analysis. The extraperitoneal approach offers the benefit of avoiding the peritoneal cavity, minimizing unnecessary contact with the organs within it, and allows surgery to be performed without Trendelenburg positioning, which has been shown to impact the hemodynamics of the patient [29]. Furthermore, Chavali et al. previously demonstrated that SP extraperitoneal RARP is associated with reduced morbidity compared to MP transperitoneal RARP, facilitating the transition to outpatient surgery [30]. Given the imbalance between the extraperitoneal and transperitoneal approaches in our MP and SP cohorts, we believe that the surgical approach might have significantly influenced the incidence of postoperative complications and SDD incidence.

Moreover, LOS was significantly correlated only with postoperative complications at 30 days, which indicates that for each hour of hospitalization, the risk of developing a complication within 30 days increased by 2%. However, this association was weaker than the effect of the type of procedure and was close to the significance threshold (*p* = 0.046). LOS was significantly improved for SP procedures in the elderly group, with the median LOS being less than 24 h of hospitalization for SP procedures. Furthermore, SDD was significantly higher for SP procedures than for MP procedures in both groups. Our findings align with those of the current literature, which demonstrated that same-day discharge procedures are safe and feasible in urologic robotic procedures, especially with the SP platform, as enhanced by the extraperitoneal approach [30]. Furthermore, our results not only support this notion but also suggest that age should not preclude patients from being eligible for same-day discharge.

In conclusion, SP-RARP is a safe and feasible option for elderly and frail patients, associated with fewer short-term postoperative complications compared to MP-RARP. This should be considered when recommending surgical treatments for elderly and fragile patients, as it may result in better postoperative outcomes.

However, this study is not without limitations. First, this is a retrospective study, and both selection bias and confounding factors could have been introduced. Secondly, procedures were performed by four different surgeons. Despite the fact that all surgeons were highly experienced in robotic surgery, this could introduce variability in outcomes. Additionally, as this is a retrospective study, all SP cases were included, including those performed early in the learning curve. This could influence outcomes, as Pellegrino et al. have highlighted the challenges of achieving proficiency in SP-RARP [31]. Therefore, it is possible that, accounting for surgeons’ experience, postoperative complications might further decrease once the learning curve is considered. Furthermore, future research examining postoperative outcomes in elderly patients should also investigate functional outcomes comparing elderly and non-elderly patients, as well as assess the costs associated with different robotic platforms. For these reasons, further studies are required to corroborate our conclusions.

## 5. Conclusions

In conclusion, SP-RARP is a safe and feasible option for elderly and frail patients, associated with fewer short-term postoperative complications compared to MP-RARP. This should be considered when recommending surgical treatments for elderly and fragile patients, as it may result in better postoperative outcomes. Furthermore, age should not preclude SDD procedures if performed with the SP technique.

## Figures and Tables

**Figure 1 cancers-17-01857-f001:**
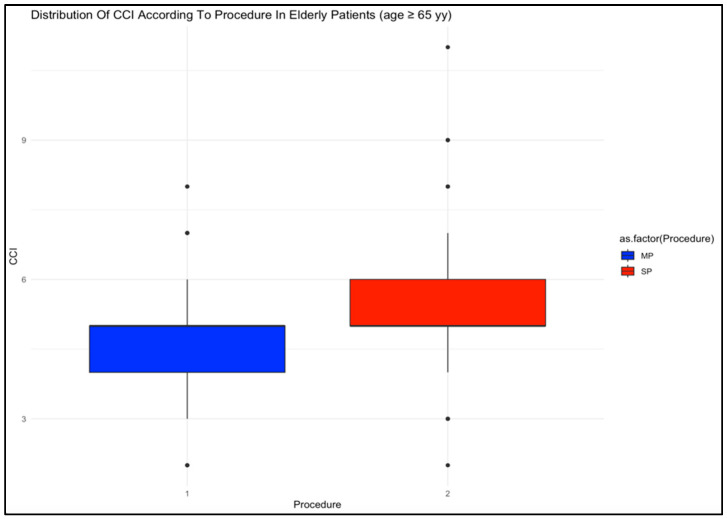
Distribution of CCI according to procedure in elderly patients. CCI: Charlson Comorbidity Index; MP: multi-port; SP: single port.

**Figure 2 cancers-17-01857-f002:**
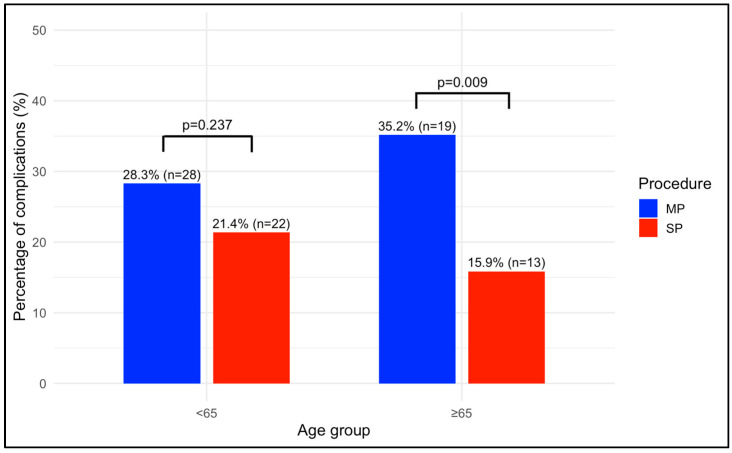
Rate of Postoperative Complications at 30 Days. *p* values indicate the differences between procedures. MP: multi-port; SP: single port.

**Figure 3 cancers-17-01857-f003:**
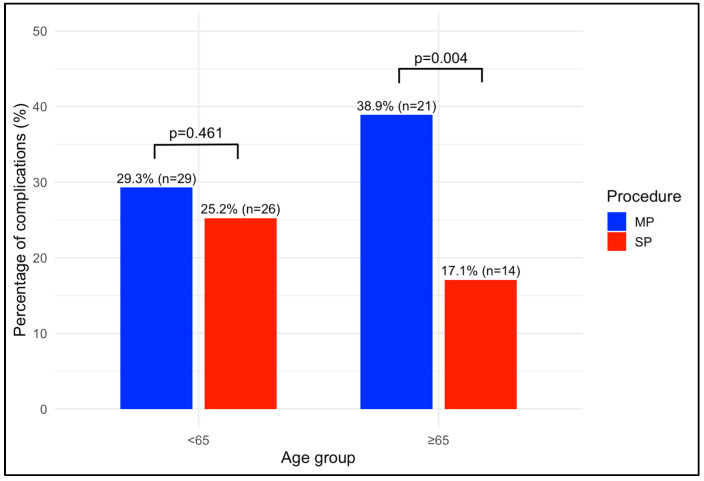
Rate of Postoperative Complications at 90 Days. *p* values indicate the differences between procedures. MP: multi-port; SP: single port.

**Table 1 cancers-17-01857-t001:** Preoperative characteristics. MP: multi-port; SP: single port; BMI: body mass index; ASA: American Society of Anesthesiologists score; CCI: Charlson Comorbidity Index; ISUP: International Society of Urological Pathology.

Variable	GROUP A Age < 65			GROUP B Age ≥ 65			*p* Value Within the MP Group According to Age Group	*p* Value Within the SP Group According to Age Group
	MP	SP	*p* Value	MP	SP	*p* Value		
N° of cases, n (%)	99 (49.0)	103 (51.0)		54 (39.7)	82 (60.3)			
Age yy, median (IQR)	60 (6)	60 (6)	0.219	69 (4)	69 (5)	0.504	**<0.001**	**<0.001**
BMI, Kg/m^2^	29.9 (7.54)	28.1 (8.13)	0.930	27.9 (5.58)	28.1 (6.70)	0.361	**0.025**	0.194
ASA, n (%)			0.254			0.127	0.128	0.647
1	1 (1.0)	0 (0.0)		0 (0.0)	0 (0.0)			
2	44 (44.4)	49 (47.6)		17 (31.5)	38 (46.3)			
3	51 (51.5)	54 (52.4)		37 (68.5)	44 (53.7)			
4	3 (3.1)	0 (0.0)		0 (0.0)	1 (1.2)			
CCI n, median (IQR)	3 (2)	4 (2)	**<0.001**	5 (1)	5 (1)	**0.001**	**<0.001**	**<0.001**
D’Amico risk classification, n (%)			0.612			0.055	0.076	0.840
Low	8 (8.1)	13 (12.6)		0 (0.0)	8 (9.8)			
Intermediate	80 (80.8)	79 (76.7)		46 (85.2)	65 (79.2)			
High	11 (11.1)	11 (10.7)		8 (14.8)	9 (11.0)			
Race, n (%)			0.178			0.151	0.571	0.132
African American	65 (65.6)	64 (62.1)		31 (57.4)	41 (50.0)			
Caucasian	19 (19.2)	15 (14.6)		12 (22.2)	16 (19.5)			
Hispanic	8 (8.2)	20 (19.4)		5 (9.3)	15 (18.3)			
Asian	1 (1.0)	1 (1.0)		2 (3.6)	2 (2.4)			
Other	6 (6.0)	3 (2.9)		4 (7.5)	8 (9.8)			
Smoking, n (%)			**<0.001**			0.107	0.510	0.205
No	35 (35.4)	66 (64.1)		22 (40.7)	45 (54.9)			
Yes	64 (64.6)	37 (35.9)		32 (59.3)	37 (45.1)			
Substance Abuse, n (%)			0.730			0.573	0.492	0.657
no	66 (66.7)	71 (68.9)		33 (61.1)	54 (65.9)			
yes	33 (33.3)	32 (31.1)		21 (38.9)	28 (34.1)			
Hypertension, n (%)			0.743			0.973	0.227	0.258
No	27 (27.3)	26 (25.2)		10 (18.5)	15 (18.3)			
Yes	72 (72.2)	77 (74.8)		44 (81.5)	67 (81.7)			
Hypertension therapy, n (%)			0.862			0.959	0.283	0.310
No	28 (28.3)	28 (27.2)		11 (20.4)	17 (20.7)			
Yes	71 (71.7)	75 (72.8)		43 (79.6)	65 (79.3)			
Anticoagulation or Antiplatelet therapy, n (%)			0.266			0.362	**0.005**	**0.002**
No	67 (67.7)	77 (74.8)		24 (44.4)	43 (52.4)			
Yes	32 (32.3)	26 (25.2)		30 (55.6)	39 (47.6)			
PSA ng/mL, median (IQR)	7.5 (6.02)	7.5 (7.43)	0.350	7.5 (7.43)	7.4 (7.72)	0.179	0.997	0.639
ISUP, n (%)			0.770			0.072	0.061	0.851
1	19 (19.2)	15 (14.6)		1 (1.9)	12 (14.6)			
2	41 (41.4)	48 (46.6)		21 (38.8)	33 (40.2)			
3	14 (14.1)	17 (16.5)		13 (24.1)	18 (22.0)			
4	19 (19.2)	16 (15.5)		13 (24.1)	11 (13.4)			
5	6 (6.1)	7 (6.8)		6 (11.1)	8 (9.8)			
cT stage, n (%)			0.547			0.649	0.246	0.094
T1	78 (78.8)	86 (83.5)		37 (68.5)	63 (76.8)			
T2	13 (13.1)	13 (12.6)		8 (14.8)	8 (9.8)			
T3	8 (8.1)	4 (3.9)		9 (16.7)	11 (13.4)			
cN stage, n (%)			0.326			0.412	0.898	0.864
N0/Nx	99 (100)	102 (99)		54 (100)	81 (98.8)			
N1	0 (0.0)	1 (1.0)		0 (0.0)	1 (1.2)			

Bold has been used to highlight the significant *p* values (<0.05) and the variables names.

**Table 2 cancers-17-01857-t002:** Intraoperative and Postoperative characteristics. MP: multi-port; SP: single port; LOS: length of stay; PONV: postoperative nausea and vomiting; EBL: estimated blood loss; CD: Clavien–Dindo classification; * log-rank test.

Variable	Group A Age < 65			Group B Age ≥ 65			*p* Value Within the MP Group According to Age Group	*p* Value Within the SP Group According to Age Group
	MP	SP	*p* Value	MP	SP	*p* Value		
N° of cases, n (%)	99 (49.0)	103 (51.0)		54 (39.7)	82 (60.3)			
Access, n (%)			**<0.001**			**<0.001**	1.000	0.818
Transperitoneal	99 (100)	36 (35.0)		54 (100)	30 (36.6)			
Extraperitoneal	0 (0.0)	67 (65.0)		0 (0.0)	52 (63.4)			
Operative Time min, median (IQR)	288 (83.5)	241 (56.0)	**<0.001**	290 (74.0)	256 (72.3)	**<0.001**	0.084	0.098
Lymphadenectomy, n (%)			**0.026**			**<0.001**	**0.024**	0.846
No	24 (24.2)	40 (38.8)		5 (9.3)	33 (40.2)			
Yes	75 (75.8)	63 (61.2)		49 (90.7)	49 (59.8)			
EBL ml, median (IQR)	150 (150)	100 (150)	**0.008**	150 (188)	100 (180)	0.052	0.352	0.210
Intraoperative complications, n (%)			0.584			0.381	0.251	0.074
No	98 (99.0)	101 (98.1)		52 (96.3)	76 (92.7)			
Yes	1 (1.0)	2 (1.9)		2 (3.7)	6 (7.3)			
LOS hours, median (IQR)	34 (22)	16 (21)	0.785	35 (17)	18 (20.8)	**0.002**	0.389	0.332
SDD, n (%)			**<0.001**			**<0.001**	0.643	0.477
No	78 (78.8)	40 (38.8)		45 (83.3)	37 (45.1)			
Yes	21 (21.2)	63 (61.2)		9 (16.7)	45 (54.9)			
PONV, n (%)						**0.040**	0.776	0.112
No	70 (70.7)	98 (95.1)		41 (75.9)	73 (89.1)			
Yes	29 (29.3)	5 (4.9)		13 (24.1)	9 (10.9)			
30 days postoperative complications, n (%)			0.237			**0.009**	0.398	0.342
No	71 (71.7)	81 (78.6)		35 (64.8)	69 (84.1)			
Yes	28 (28.3)	22 (21.4)		19 (35.2)	13 (15.9)			
30 days CD, n (%)			0.222			0.557	0.590	0.833
1	11 (39.3)	6 (27.4)		6 (31.7)	3 (23.1)			
2	10 (35.7)	14 (63.6)		7 (36.8)	8 (61.5)			
3	5 (17.9)	1 (4.5)		5 (26.3)	2 (15.4)			
4	2 (7.1)	1 (4.5)		0 (0.0)	0 (0.0)			
5	0 (0.0)	0 (0.0)		1 (5.2)	0 (0.0)			
90 days postoperative complications, n (%)			0.461			**0.004**	0.260	0.180
No	70 (70.7)	77 (74.8)		33 (61.1)	68 (82.9)			
Yes	29 (29.3)	26 (25.2)		21 (38.9)	14 (17.1)			
90 days CD, n (%)			0.387			0.511	0.387	0.511
1	11 (37.9)	7 (26.9)		7 (33.3)	4 (28.6)			
2	11 (37.9)	14 (53.8)		8 (38.1)	8 (57.1)			
3	5 (17.3)	3 (11.6)		5 (23.8)	2 (14.3)			
4	2 (6.9)	2 (7.7)		0 (0.0)	0 (0.0)			
5	0 (0.0)	0 (0.0)		1 (4.8)	0 (0.0)			
ISUP, n (%)						0.051	0.083	0.105
1	9 (9.1)	5 (4.9)		0 (0.0)	4 (4.9)			
2	49 (49.4)	61 (59.2)		27 (50.0)	35 (42.7)			
3	27 (27.3)	22 (21.4)		14 (25.9)	25 (30.5)			
4	6 (6.1)	6 (5.8)		3 (5.5)	12 (14.6)			
5	8 (8.1)	9 (8.7)		10 (18.6)	6 (7.3)			
pT stage, n (%)			0.725			0.282	0.622	0.431
T1	2 (2.1)	1 (1.0)		0 (0.0)	2 (2.4)			
T2	51 (51.5)	58 (56.3)		32 (59.3)	39 (47.6)			
T3	45 (45.4)	43 (41.7)		22 (40.7)	41 (50.0)			
T4	1 (1.0)	1 (1.0)		0 (0.0)	0 (0.0)			
pN stage, n (%)			0.313			0.074	0.561	0.530
N0/Nx	89 (89.9)	90 (87.4)		50 (92.5)	69 (84.1)			
N1	10 (10.1)	13 (12.6)		4 (7.4)	13 (15.9)			
Biochemical recurrence, n (%)			0.203 *			**0.040 ***	0.856 *	0.175 *
No	76 (76.8)	86 (83.5)		42 (77.8)	74 (90.2)			
Yes	23 (23.2)	17 (16.5)		12 (22.2)	8 (9.8)			

Bold has been used to highlight the significant *p* values (<0.05) and the variables names.

**Table 3 cancers-17-01857-t003:** Multivariable logistic regression analysis predicting postoperative complications at 30 days. SP: single port; LOS: length of stay; CCI: Charlson Comorbidity Index.

Logistic Regression for Postoperative Complications at 30 Days in Patients of Group B			
Variable	Odds Ratio	95% CI	*p* Value
Procedure (SP)	0.41	0.15, 0.97	**0.027**
LOS	1.02	1.01, 1.13	**0.046**
Access (Retroperitoneal)	0.45	0.23, 1.14	0.082
Intercept	0.23	0.02, 0.68	**0.001**

Bold has been used to highlight the significant *p* values (<0.05) and the variables names.

**Table 4 cancers-17-01857-t004:** Multivariable logistic regression analysis predicting postoperative complications at 90 days. SP: single port; LOS: length of stay; PONV: postoperative nausea and vomiting; CCI: Charlson Comorbidity Index.

Logistic Regression for Postoperative Complications at 90 Days in Patients of Group B			
Variable	Odds Ratio	95% CI	*p* Value
Procedure (SP)	0.38	0.17, 0.88	**0.024**
Lymphadenectomy	1.56	0.55, 4.44	0.405
LOS	1.01	0.99, 1.02	0.217
Access (Retroperitoneal)	0.52	0.21, 1.12	0.087
Intercept	0.17	0.03, 0.86	**0.007**

Bold has been used to highlight the significant *p* values (<0.05) and the variables names.

## Data Availability

The data presented in this study are available on request from the corresponding author.

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
