# Peer review of "Single-Port Versus Multi-Port Robotic Radical Prostatectomy in Elderly Patients"

_cancers, 2025, doi:10.3390/cancers17111857_

Round 1

Reviewer 1 Report

Comments and Suggestions for Authors

This is a retrospective study comparing single-port (SP) versus multi-port (MP) robotic-assisted radical prostatectomy (RARP) with a focus on elderly patients. The study addresses an important clinical question regarding the optimal surgical approach for this patient population, who often have comorbidities and may benefit from less invasive procedures. The authors present data suggesting that SP-RARP is associated with fewer short-term postoperative complications and facilitates same-day discharge in elderly patients compared to MP-RARP. While the study has several strengths, including a reasonable sample size and relevant clinical outcomes, there are also some limitations that need to be addressed.  
1.    The use of age 65 as a cutoff is common, but was this based on clinical or statistical reasoning specific to this cohort? Could the analysis be enhanced with further stratification (e.g., 65–75 vs. >75 years)?
2.    The conclusion suggests that SP RARP is preferable for elderly patients. Given the higher CCI in the SP group, could this recommendation apply equally to frail patients with multiple comorbidities?
3.    Were there any specific criteria for selecting SP versus MP procedures? For example, did surgeon preference or patient characteristics influence the choice of approach?
4.    Early SP cases may reflect the surgeons' learning curve. How many of the SP cases were performed in the early phase of implementation? Was there any temporal trend in outcomes?
5.    The manuscript acknowledges variability due to multiple surgeons and the learning curve for SP procedures. Were outcomes analyzed by surgeon or by year to account for this variability?
6.    The manuscript states that all MP procedures were transperitoneal, while SP procedures included both transperitoneal and extraperitoneal approaches. Could the choice of approach (extraperitoneal vs. transperitoneal) in the SP group have confounded the results, particularly regarding complications and recovery?
7.    The study period spans 2018–2023. Were there any changes in surgical techniques or perioperative protocols during this time that could have affected the results?
8.    Were any cost comparisons performed between SP and MP RARP (operative time, LOS, equipment usage)?
9.    Were any functional outcomes (continence, erectile function) evaluated postoperatively? If not, do you plan to assess these in future studies?
10.    The retrospective nature of the study introduces the potential for selection bias and confounding. The authors should acknowledge this limitation more explicitly and discuss potential sources of bias.

Author Response

  1.    The use of age 65 as a cutoff is common, but was this based on clinical or statistical reasoning specific to this cohort? Could the analysis be enhanced with further stratification (e.g., 65–75 vs. >75 years)?
    A: Thank you for your comment. Age 65 was chosen as it is commonly used as a standard cutoff in existing literature. We agree that further stratifying elderly patients into subgroups would be valuable. Unfortunately, initial attempts at subgroup analysis resulted in sample sizes too small for meaningful statistical analysis, which would have weakened the study.

  2.    The conclusion suggests that SP RARP is preferable for elderly patients. Given the higher CCI in the SP group, could this recommendation apply equally to frail patients with multiple comorbidities?
    A: Thank you for your insightful comment. We agree that a higher CCI does not negatively impact the outcomes for elderly patients undergoing SP prostatectomy. This clarification has now been included in the conclusion section.

  3.    Were there any specific criteria for selecting SP versus MP procedures? For example, did surgeon preference or patient characteristics influence the choice of approach?
    A: Thank you for your question. No specific patient criteria determined the choice of robotic approach. The decision between MP and SP approaches was based solely on surgeon preference and availability of the robotic platform.

  4.    Early SP cases may reflect the surgeons' learning curve. How many of the SP cases were performed in the early phase of implementation? Was there any temporal trend in outcomes?
    A: Thank you for your comment. As mentioned in the manuscript, we evaluated all consecutive SP RARP cases since its introduction into our clinical practice. This was essential since, as demonstrated by Pellegrino et al. (reference 21), the risk of postoperative complications decreases significantly after the first 150 cases. Although a temporal trend was not specifically evaluated, we recognize the importance of this aspect and suggest that further research may be beneficial.

  5.    The manuscript acknowledges variability due to multiple surgeons and the learning curve for SP procedures. Were outcomes analyzed by surgeon or by year to account for this variability?
    A: Thank you for your comment. We acknowledge that variability in outcomes (30-day and 90-day postoperative complications) could have been influenced by differences among surgeons performing radical prostatectomy, as well as by the early-stage adoption of the new robotic platform compared to the well-established MP approach. However, outcomes were not analyzed by individual surgeons or by year, as this was beyond the scope of our study.

  6.  The manuscript states that all MP procedures were transperitoneal, while SP procedures included both transperitoneal and extraperitoneal approaches. Could the choice of approach (extraperitoneal vs. transperitoneal) in the SP group have confounded the results, particularly regarding complications and recovery?
    A: Thank you for your insightful comment. As stated in the manuscript, the retroperitoneal approach may offer advantages such as facilitating same-day discharge and potentially reducing postoperative complications. However, our logistic regression analysis did not find the surgical approach to be a statistically significant predictor of postoperative complications. Despite this, we acknowledge that the imbalance in surgical approaches between MP and SP could have influenced our results, and this point has been clarified further in the manuscript.

  7.    The study period spans 2018–2023. Were there any changes in surgical techniques or perioperative protocols during this time that could have affected the results?
    A: Thank you for your comment. No changes in surgical techniques or perioperative protocols occurred during the study period. This clarification has now been explicitly added to the materials and methods section.

  8.    Were any cost comparisons performed between SP and MP RARP (operative time, LOS, equipment usage)?

A: Thank you for your comment. A cost analysis was not performed, as it was outside the scope of our current study objectives. Nevertheless, we recognize the importance of such an analysis, and future studies addressing cost-effectiveness between SP and MP platforms will be essential. This point has been included in the conclusion.

  1.    Were any functional outcomes (continence, erectile function) evaluated postoperatively? If not, do you plan to assess these in future studies?
    A: Thank you for your comment. Functional outcomes were not evaluated postoperatively, as our study specifically aimed to assess early postoperative complications related directly to the surgical procedure. We agree, however, that evaluating functional outcomes is crucial, especially in the elderly population, and future studies should certainly address this aspect.

  2.    The retrospective nature of the study introduces the potential for selection bias and confounding. The authors should acknowledge this limitation more explicitly and discuss potential sources of bias.
    A: Thank you for your comment. We have explicitly acknowledged and elaborated upon these limitations and potential sources of bias in the revised manuscript.

Reviewer 2 Report

Comments and Suggestions for Authors

This is a well-structured and clearly written retrospective study comparing single-port (SP) and multi-port (MP) robotic radical prostatectomy in elderly patients. The topic is clinically relevant, and the large cohort and subgroup analyses are valuable. The methodology and statistical approach are appropriate, and the reporting of outcomes is generally sound. The paper would benefit from some clarifications, especially regarding potential confounders and interpretation of results. Overall, the data are well presented and the conclusions mostly supported, though a few are slightly overstated and could be more cautiously phrased.

Comments

  1. The manuscript refers to its novelty in comparing SP and MP robotic prostatectomy specifically in elderly patients. This should be more clearly emphasized in both the introduction and discussion. The authors may wish to contrast their approach with existing literature—particularly the study by Haberal et al. (2025), which focused on SP in seniors but did not offer a comparative design.

  2. Since all MP procedures were transperitoneal and a significant portion of SP procedures were extraperitoneal, the surgical approach itself may have influenced outcomes such as complications or length of stay. This should be more explicitly acknowledged as a limitation, even if not statistically significant in multivariable analysis. If feasible, consider including a subgroup analysis (SP transperitoneal vs extraperitoneal) for this or future studies.

  3. LOS is reported in hours, and the concept of same-day discharge (SDD) is mentioned, but specific institutional criteria for SDD are not described. Providing a brief description of these criteria (e.g., catheter removal, stable vitals, ambulation) would improve reproducibility and interpretation.

  4. Consider providing subparagraphs for patient population and endpoint of the studies in the methods section to improve readability and access to importat information.

Author Response

  1. The manuscript refers to its novelty in comparing SP and MP robotic prostatectomy specifically in elderly patients. This should be more clearly emphasized in both the introduction and discussion. The authors may wish to contrast their approach with existing literature—particularly the study by Haberal et al. (2025), which focused on SP in seniors but did not offer a comparative design.

A: Thank you for your valuable comment. We have now provided a clearer emphasis on the novelty of comparing SP and MP robotic prostatectomy specifically in elderly patients in both the introduction and discussion sections, also highlighting contrasts with existing literature, such as the study by Haberal et al. (2025).

  1. Since all MP procedures were transperitoneal and a significant portion of SP procedures were extraperitoneal, the surgical approach itself may have influenced outcomes such as complications or length of stay. This should be more explicitly acknowledged as a limitation, even if not statistically significant in multivariable analysis. If feasible, consider including a subgroup analysis (SP transperitoneal vs extraperitoneal) for this or future studies.
    A: Thank you for your insightful comment. Following your recommendation, we have explicitly acknowledged this limitation in the discussion section, clearly highlighting the differences between MP transperitoneal and SP extraperitoneal procedures. We have also referenced the study by Chavali et al., which addresses this concern.

  2. LOS is reported in hours, and the concept of same-day discharge (SDD) is mentioned, but specific institutional criteria for SDD are not described. Providing a brief description of these criteria (e.g., catheter removal, stable vitals, ambulation) would improve reproducibility and interpretation.
    A: Thank you for your suggestion. An explicit description of the institutional criteria for same-day discharge has now been provided in the materials and methods section to enhance reproducibility and clarity.

  3. Consider providing subparagraphs for patient population and endpoint of the studies in the methods section to improve readability and access to importat information.
    A: Thank you for your suggestion. Subparagraphs have been incorporated into the methods section to clearly delineate the patient population and study endpoints, thus enhancing readability and accessibility.

Reviewer 3 Report

Comments and Suggestions for Authors

This is a non-randomised case control series of MP vs SP prostatectomy in a cohort of men and sub-divided by age ( dichotomised at 65 years).

The authors should recognise the effect of case selection in this publication and discuss the implications of such a bias towards the outcomes that they report.  These include the reasons behind selecting patients to either SP or MP, learning curves of surgeons each modality etc.   In addition, the rate of clavien Dindo 3 and above complications look more prominent than contemporary RARP series.  It would help for the authors to share information on details of CD 3 and above complications so as to allow the reviewers to understand the surgical outcomes comprehensively.

Author Response

The authors should recognise the effect of case selection in this publication and discuss the implications of such a bias towards the outcomes that they report.  These include the reasons behind selecting patients to either SP or MP, learning curves of surgeons each modality etc.   In addition, the rate of clavien Dindo 3 and above complications look more prominent than contemporary RARP series.  It would help for the authors to share information on details of CD 3 and above complications so as to allow the reviewers to understand the surgical outcomes comprehensively.

A: Thank you for your comment. The study's limitations have been reviewed and expanded. Additionally, a detailed explanation of Clavien-Dindo grade 3 and higher complications has been added to the results section.

Round 2

Reviewer 3 Report

Comments and Suggestions for Authors

none

Author Response

Thank you